# Accounting for Weather Variability in Farm Management Resource Allocation in Northern Ghana: An Integrated Modeling Approach

Opeyemi Obafemi Adelesi [1], Yean-Uk Kim [1], Heidi Webber [1,*], Peter Zander [1], Johannes Schuler [1], Seyed-Ali Hosseini-Yekani [1], Dilys Sefakor MacCarthy [2], Alhassan Lansah Abdulai [3], Karin van der Wiel [4], Pierre C. Sibiry Traore [5,6] and Samuel Godfried Kwasi Adiku [7]

[1] Leibniz Centre for Agricultural Landscape Research (ZALF), 15374 Müncheberg, Germany; adelesi.opeyemi_obafemi@zalf.de (O.O.A.)

[2] Soil and Irrigation Research Centre, School of Agriculture, University of Ghana, Accra P.O. Box LG 68, Ghana

[3] CSIR-Savanna Agricultural Research Institute, Tamale P.O. Box TL 52, Ghana

[4] Royal Netherlands Meteorological Institute, 3731 GA De Bilt, The Netherlands

[5] ICRISAT-Senegal, Dakar P.O. Box 24365, Senegal

[6] Manobi Africa PLC, Dakar P.O. Box 25026, Senegal

[7] Department of Soil Science, University of Ghana, Accra P.O. Box LG 245, Ghana

* Correspondence: webber@zalf.de

**Abstract:** Smallholder farmers in Northern Ghana face challenges due to weather variability and market volatility, hindering their ability to invest in sustainable intensification options. Modeling can help understand the relationships between productivity, environmental, and economical aspects, but few models have explored the effects of weather variability on crop management and resource allocation. This study introduces an integrated modeling approach to optimize resource allocation for smallholder mixed crop and livestock farming systems in Northern Ghana. The model combines a process-based crop model, farm simulation model, and annual optimization model. Crop model simulations are driven by a large ensemble of weather time series for two scenarios: good and bad weather. The model accounts for the effects of climate risks on farm management decisions, which can help in supporting investments in sustainable intensification practices, thereby bringing smallholder farmers out of poverty traps. The model was simulated for three different farm types represented in the region. The results suggest that farmers could increase their income by allocating more than 80% of their land to cash crops such as rice, groundnut, and soybeans. The optimized cropping patterns have an over 50% probability of increasing farm income, particularly under bad weather scenarios, compared with current cropping systems.

**Keywords:** bio-economic farm model; integrated model; weather risk; mixed cropping system; CLEM; Northern Ghana; SIMPLACE

## 1. Introduction

There is considerable pressure on smallholder farmers in sub-Saharan Africa to increase production as a means to improve their income and food security, as well as to supply food to larger and growing urban markets [1–4]. With land expansion posing an unacceptable pressure on biodiversity, sustainable intensification of agriculture is considered to be an increasingly appropriate approach. This implies intensifying agricultural production [2,5] while ensuring environmental protection, maintaining soil fertility and soil carbon, preserving biodiversity, and building the climatic resilience of farming systems [1,2,6].

Despite the considerable effort to promote the sustainable intensification of smallholder farming in semi-arid West Africa [7], success remains limited. Farmers are challenged by many constraints related to poor soil fertility, weak market infrastructure, lack of access to credit facilities, fluctuating prices, and high weather variability [8], which has

made farming very risky [3,9–15]. While farmers face many sources of risk, including price volatility and market shocks, illness of family members and livestock, and pest and disease outbreaks [15], the risks related to (extreme) weather such as heat and drought remain significant because of their frequency and magnitude being projected to increase with climate change [16–18]. Together, the wide range of risks and uncertainties faced by farmers in the region are key reasons for them not investing in improved farming practices [12].

Mixed crop—livestock farmers in Northern Ghana have adopted several management strategies on their farms to minimize the impact of these risks. Farmers may choose more drought-resistant crops and varieties, change planting dates, and/or plant a mixture of crops and varieties [19]. However, the benefits of these risk-reducing practices are unlikely to be sufficient in years with severe yield failures due to drought [20,21] or pests, which is the reason behind many of the current efforts to promote insurance products. Many farmers in the greater West African Sudan Savannah already use livestock as informal insurance in the case of insufficient income from cropping. In addition to providing traction and food, animals can be sold in years with crop yield failure to overcome cash shortages [22]. As such, crop residues are often preferably used as fodder instead of being incorporated into soils to improve soil fertility [20,23,24]. On the other hand, formal insurance products remain unattractive to many farmers for a number of reasons [25], including that their cost in good years may preclude other investments. Comprehensively considered, managing risk in farming will likely need elements of risk reduction, risk transfer, savings and smart risk-taking, among other factors [26], and the strategy depends on the specifics of the farms, agro-ecology, and institutional and market factors. In the context of an increasingly variable and extreme climate, it is important to better assess which combination of risk management options best serves the livelihood objectives of smallholder farmers.

Given the host of the soil—crop—weather—animal—market complexity of a farming system, system analysis tools are helpful for complementing experimentation and for exploring a wide range of options and conditions. System modeling that integrates biophysical and economic models is one option for investigating complex systems by capturing crop and livestock production, soil fertility-related processes, on-farm and off-farm labor availability, and livestock feeds in addition to socio-economic behavior of farmers [4,27]. Such models can help to better understand farmers' behavior and the effects on resource allocation [28]. They can also allow for the evaluation of likely outcomes of proposed interventions and for the identification of interactions among components across scales and levels [29]. This is evident in models such as the one described by [29] and often include linear programming, system dynamics, and agent-based modeling. Several models could also be combined in order to explain the interactions among complex socio-economic, bio-physical, and socio-ecological processes, which are often referred to as bio-economic farm models [29–32].

There are examples of modeling approaches that are intended to capture the effects of weather, price, and production risks on different farm household components, such as consumption [4,8,10]. However, to the best of our knowledge, none have yet included feedback on crop and soil management due to altered farm-level decisions. This is considered to be crucial in assessing how weather (and other) risks affect crop management and the suitability of different risk management strategies. It is particularly important when considering the implications for longer-term sustainability outcomes with respect to environmental variables (soil organic carbon and biodiversity). Furthermore, while the impact of shocks and risk management options can be evaluated through scenario analyses [33], models that can simultaneously evaluate farmers' response to weather shocks and the subsequent impacts on farm assets such as livestock or natural capital are still lacking. Furthermore, previous studies have focused on optimizing production in the face of risks over a typical planning period, with few attempts having been made to account for the effects of shocks on farm trajectories [34].

In this context, the objective in this study was to develop an integrated modeling approach that captures the probability of changes in assets, incomes, or resource allocation

in response to degrees of weather variability and price volatility for smallholder mixed farming systems in Northern Region, Ghana. To this end, we combined a crop model, farm simulation model, and optimization model to assess the effects of weather- and market-related variability in the optimization of resource allocation. Two weather scenarios (relatively good and bad) constructed from large ensemble climate modeling were used to force the integrated model.

## 2. Materials and Methods

### 2.1. Study Area

The study was based on data from the Northern Region, Ghana, which is located in the Guinea Savannah ecological zone. Up until a recent re-demarcation into new administrative regions, the zone comprised three main regions, namely the Northern Region, Upper West Region and Upper East Region. The Northern Region has a population of approximately 2.5 million inhabitants and has a large rural farming population [35]. It is characterized by a single rainy season occurring between May and October, with an annual rainfall between 750 mm and 1050 mm and an annual mean temperature between 22.4 °C and 33.9 °C. There is typically a prolonged dry season between November and March/April, and the impacts of climate change (frequent floods, drought, and bush fires) are already pronounced in the region [3,36,37]. The main food crops grown in the Northern Region include maize, rice, sorghum, millet, cowpea, and groundnut. Cash crops such as soybean, tomato, onion, and leafy vegetables are grown in the area, while livestock reared in the area include cattle and small ruminants such as goats and sheep [38].

### 2.2. Sampling Technique and Data Collection

An initial household survey of 700 households was carried out during the 2020 agricultural season across the Upper West Region, Upper East Region, and Northern Region. The data were filtered for the Northern Region households only, which reduced the dataset to 378 households located in the Tolon, Savelugu, and Mion districts. The data were used to develop a farm typology based on socio-economic characteristics such as age, sex, and farm resource endowments (mainly land and herd size). The data were clustered into three farm types with different resource endowments (described in the following section). A second in-depth survey was carried out before the 2022 planting season to obtain detailed crop and livestock production data, which were used in the modeling exercise. Respondents were randomly selected, resulting in a total of 45 households distributed equally between our three farm types. Data were obtained with the aid of structured questionnaires added to the JotBi app developed by CGIAR CASCAID [39]. Historical crop price distributions were obtained from the Ghanaian Ministry of Food and Agriculture reports [40,41] and adjusted for inflation using the producer's price index data obtained from [42].

### 2.3. Formation of Farm Typology

The selection of variables for the typology was performed based on the literature, expert opinion, context, and local context [43–45]. Ten variables (Table 1) that best classify the households based on income and resource endowments in the dataset were used to cluster the farmers. The diversity among farmers was defined based on household characteristics and resource endowment. The dataset was checked for missing data and outliers, and these were controlled for through imputation and list-wise deletion techniques as proposed by [43,46]. Of the 378 households, 340 were retained for analysis, and the farm households were clustered into 3 farm types, comprising 61 low-resource-endowed farms, 181 medium-resource-endowed farms, and 98 high-resource-endowed farms, respectively. The farm household typology was constructed using a principal component analysis (PCA) on the 340 households with a factor analysis on mixed data (FAMD) analysis. The "FAMD" function within the "FactoMineR" package in R software (Version 4.0.2.) was used [47,48] as it is best suited for both continuous and categorical variables [49]; we also used the package to carry out hierarchical clustering to obtain the different typologies [43].

**Table 1.** Summary of variables used for typologies.

| Variable | Description | Unit |
|---|---|---|
| Age | Age of the household head | years |
| Cash at hand | Cash at hand at the beginning of the season | GHS |
| Sex | Sex of the household head | - |
| Household size | Number of individuals in the household | - |
| Herd size | Total herd size | - |
| Input costs | Total cost of production inputs | GHS/year |
| Land size | Total land size | ha |
| Main crop | Main crop cultivated by farmers | - |
| Non-farm income | Annual household off farm income | GHS/year |
| Total annual income | Total annual household income | GHS/year |

GHS is the Ghanaian Cedi, which is the official Ghanaian currency.

*2.4. Meteorological Forcing Data*

To assess the model's response to weather variability and risk, large ensemble climate modeling data were used. Large ensemble climate modeling is a technique that produces many different weather realizations for a given period and state of the climate, effectively sampling the whole distribution of possible weather [50,51]. This means that we can directly assess the risks of extreme events by running the simulated weather data through the integrated bio-economic model.

The large ensemble dataset used here contains 2000 years of present-day weather conditions, which were generated using EC-Earth global climate model data [52]. EC-Earth is an Earth system model representing atmosphere, ocean, land, and ice conditions. The ensemble consists of 400 members, each consisting of a 5-year simulation of a period representing the present-day global climate (as observed in 2011–2015). Details on the large ensemble experimental set-up can be found in [50]. The data were previously used for crop modeling studies in [8,53–55], and the data were extracted for the grid point that was nearest to the study site, Tamale (09° N and 00° W).

The modeled temperature and precipitation data were bias-corrected using station data from Nyankpala. For temperature, the daily maximum, minimum, and annual cycle were computed using a harmonic function [56]; the simulated data were then corrected by the difference between the harmonic based on the observed data and the harmonic based on the simulated data.

$$\mu(t) = a + \sum_{k=1}^{K} b_k \sin(k\omega t) + c_k \cos(k\omega t) \tag{1}$$

where

$\mu$ = Temperature (C);
$t$ = Day- of- year (1–366);
$K$ = Order of harmonic function determined using BIC (K = 4);
$\omega = 2\pi/365.25$.
$a$, $b_k$, and $c_k$ are coefficients of the harmonic function.

The precipitation was corrected on a monthly basis using a method from [54]. First, the number of precipitation days of EC-Earth data was corrected to the observed number to solve for the drizzle effect. The precipitation values of the days with precipitation amounts falling below a threshold were set to 0. The threshold was determined by matching the EC-Earth precipitation days to the observed precipitation days ($\geq$0.1 mm d$^{-1}$). After correcting the drizzle bias, the EC-Earth monthly precipitation amount was corrected by a multiplicative factor to match the observed monthly precipitation amount.

### 2.5. Modeling Approach

#### 2.5.1. Model Approach Overview

An integrated bio-economic model was developed to simulate outcomes of annual resource allocation based on the optimization of gross margins at the farm level while considering annual crop yield response to weather and management. The integrated model consisted of a process-based crop model (SIMPLACE framework) [57], a farm simulation model to account for resource flows during the year (CLEM) [58], and a newly developed optimization model. A novelty in this study was the use of the large ensemble climate forcing dataset, which allowed us to assess the probabilities of the outcomes. The three models were grouped to iteratively optimize the cropping pattern as shown in Figure 1. A flow chart of the connected models, detailing the simulation steps, is provided in Figure 2.

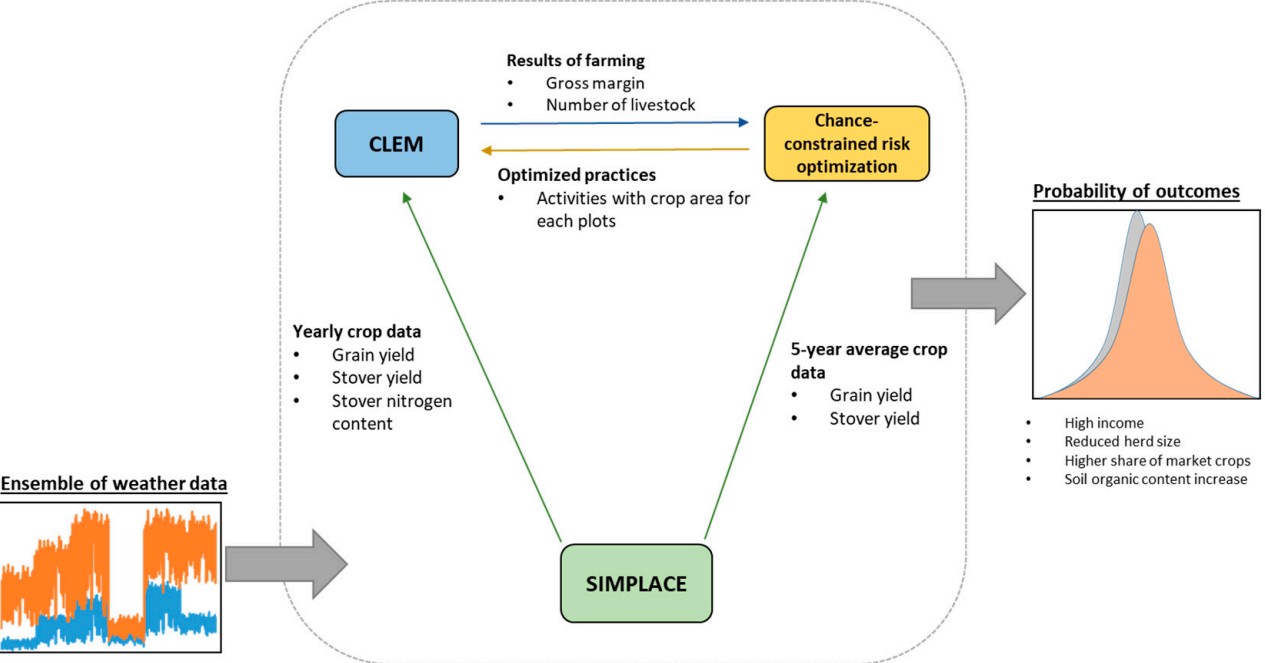

**Figure 1.** A schematic of the integrated modeling approach.

The key function of each model and the data streams between the models are as follows:

1.  SIMPLACE: simulates crop grain and biomass yield in response to weather, soil, and management. These simulations are passed to CLEM annually and to the optimization model as yield distributions across all members within a weather scenario;
2.  CLEM: simulates annual monetary and resource flows and outputs the balances of cash and herd size;
3.  Optimization model: optimizes resource allocation and the production plan for CLEM.

To explore the integrated model response under different degrees of weather variability, two weather scenarios were created from the ensemble data to represent good and bad weather conditions. This was performed by classifying climate ensemble members to either the good or bad weather scenario based on the simulated grain yields for each ensemble member. Because the grain yield level varies among the crop types (e.g., rice yield above 3000 kg ha$^{-1}$ vs. soybean yield around 1000 kg ha$^{-1}$), the relative yield was calculated for each crop type (i.e., the yield of a given year divided by the average yield in all 2000 years). Thereafter, the mean relative yield was calculated across crop types for each ensemble member (5-year simulation), and these mean yields were used to classify the ensemble members. The 30 highest and 30 lowest mean relative yield members were grouped into

the good and bad weather scenarios and were included in the simulation, as shown in Figure 2, to generate distinct weather responses in the integrated model.

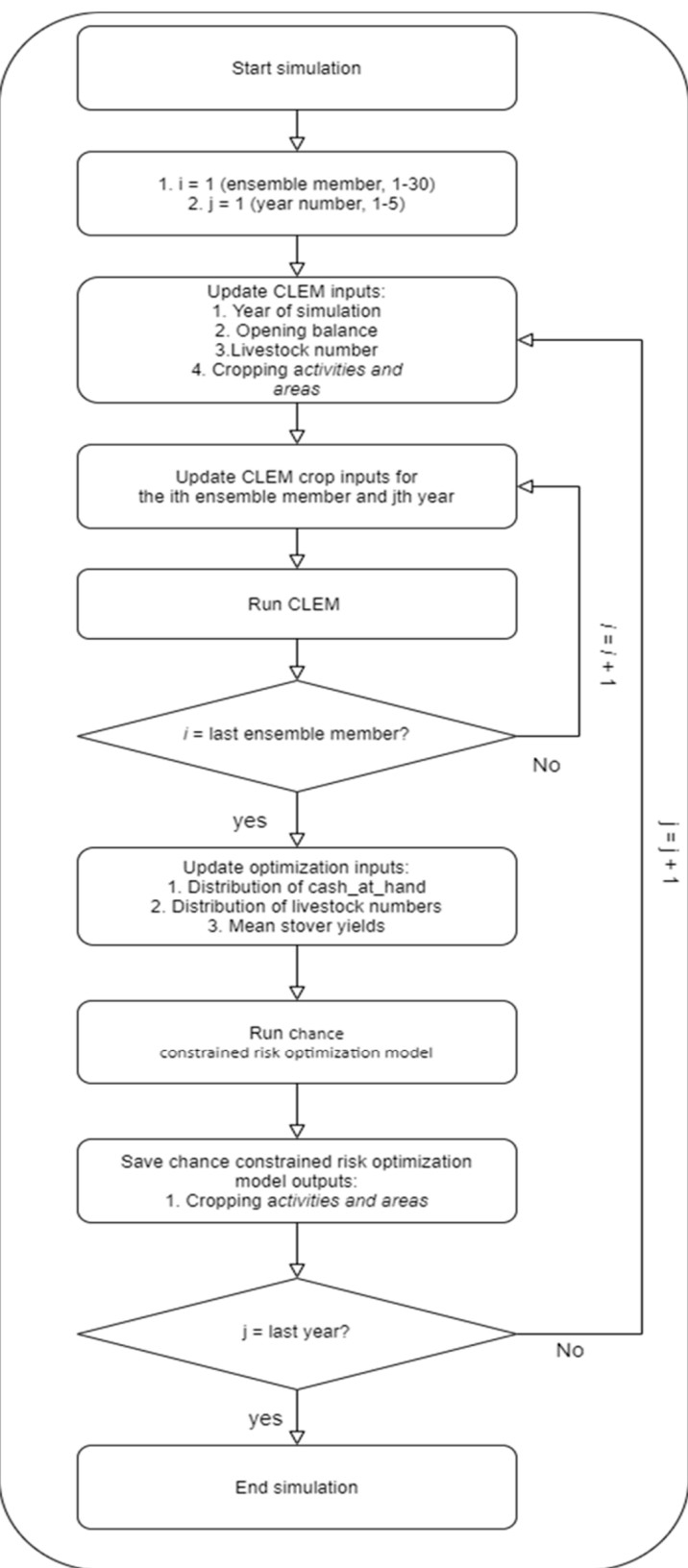

**Figure 2.** Flow chart of the integrated model.

### 2.5.2. Crop Model

The crop model simulates crop growth in response to the weather information inputted and key management choices/options [20,59,60]. These simulations are important to/for other models like the optimization model because crop yield will determine the household income and amount of food available to smallholder households in most cases, thereby affecting many household decisions. The SIMPLACE crop modeling framework (see SIMPLACE website) provided both the CLEM and optimization model a simulation of crop grain, biomass yield and nitrogen content of crops (Table 2). The above-ground growth crop module used in SIMPLACE is Lintul-5 [61], which was for this study and coupled with a modified version of Slim Water [62], various FAO-56 based modules for evapotranspiration [63], and a heat stress module [64] driven by simulated crop canopy temperature [65]. The soil inputs were generated based on the literature [66]. Crop parameters calibrated from previous studies were used [57], and crop management activities were set according to the farmers' management practices as obtained through interviews.

**Table 2.** Variable inputs-outputs among models.

| Type of Model | Base Year (Year 1) | | Subsequent Years (Year 2–5) | |
|---|---|---|---|---|
| | Variable Input | Variable Output | Variable Input | Variable Output |
| Crop Model | • Weather and soil condition<br>• Farm management | • Crop yield distributions<br>• Crop biomass | • Weather and soil condition<br>• Farm management | • Crop yield distributions<br>• Crop biomass |
| CLEM model | • Crop yield<br>• Crop biomass<br>• Initial production activities | • Net farm income<br>• Herd size | • Crop yield<br>• Crop biomass<br>• Farm production activities: crop choice | • Net farm income<br>• Herd size |
| Optimization model | • Crop yield distribution<br>• Crop biomass<br>• Farm endowments | • Farm production activities: crop choice | • Farm endowments<br>• Resource shortages<br>• Cash at hand<br>• Herd size | • Farm production activities: crop choice |

Maize crop yields were simulated at three levels of nitrogen fertilizer application (i.e., maize with low, medium, and high fertilizer application rates), referred to as maize-low, maize-medium and maize-high, respectively (see the procedure for classifying fertilizer application rates in File S1-Supplementary Materials), because maize production in the study area is constrained by fertilizer application and intensity [67]. Maize-low applied 17.5 kg N/ha of fertilizer, maize-medium applied 49.4 kg N/ha of fertilizer and maize-high applied 114 kg N/ha of fertilizer. In addition, soybean, rice, and groundnut were applied with 17.4 kg N/ha, 49 kg N/ha and 4 kg N/ha of fertilizer respectively. The simulated grain yields from the SIMPLACE model were corrected using the survey yield data and multiplicative factors for each crop type to capture the yield-reducing factors that cannot be simulated by SIMPLACE, e.g., lack of seeds, labor, herbicides, pesticides, etc.

### 2.5.3. The Crop Livestock Enterprise Model (CLEM)

The Crop Livestock Enterprise Model (CLEM) explores farm management outcomes that are subject to the available resources using the data obtained from the biophysical crop-soil model. The CLEM model is a whole farm enterprise model developed by the Commonwealth Scientific and Industrial Research Organization (CSIRO— see the CLEM website). The model combines all farm resources (i.e., labor, capital, land, equipment etc.) with farm management activities (i.e., ploughing, weeding, fertilizer application, household consumption, etc.) and provides the related balances such as net income, food in storage, etc., on a monthly basis [58]. The resources added to the model include land, finances, labor (which include both household and hired labor), animal feed store, grazing feed store, rumi-

nants, and non-ruminant animal herds. Table 2 shows all data inputs and outputs from the CLEM model and the other models. We added all farm management activities carried out by the farmers to the model, ranging from crop management to livestock management, all on-farm and off-farm labor activities, and income and expenditures, including remittances and household consumption. All of these farm resources and management activities were added for each farm type in CLEM, and the annual results include stocks of all resources and farm endowments in the simulation timeframe. Livestock are sold in the CLEM model only to meet household consumption in extreme cases where the households do not have enough money for consumption. This is performed because livestock usually serve as a store of wealth rather than serving as an item for regular trade [22] as such, households will only sell their livestock in dire situations (see File S2 in Supplementary Materials for all assumptions made in the CLEM model).

### 2.5.4. Chance-Constrained Risk Optimization Model

A whole farm non-linear mathematical programming model (The model is written in GAMS (General Algebraic Modeling System) language; full details are available from the authors upon request) was developed with the final parameterization being based on the farmers' production activities in the region. To optimize risk in bio-economic farm models, it is customary to introduce the fluctuations of crop contribution margins in the model [24,30,68]. Here, the sources of uncertainty constituting risk were the effects of weather scenarios on crop yields and the variation in herd size and cash at hand. To include these risks in the optimization model, the chance-constrained risk optimization model of [69,70] was used as:

$$Max : CE = E(GM) - RP \tag{2}$$

where

CE = Certainty equivalent of farmer's gross margin
E (GM) = Expected gross margin
RP = Farmer's risk premium
subject to:

$$\sum_{j=1}^{J} a_{ij}x_j \leq bi \ i = 1, \ldots, \ n \tag{3}$$

where

$$a_{ij} = Coefficient \ in \ the \ ith \ constraint \ for \ variable \ x_j$$

$$x_j = Level \ of \ jth \ activity$$

$$b_n (b_m) = Endowment \ of \ the \ nth \ input \ (mth \ ``non \ certain'' \ input)$$

$$Prob\left[\sum_{j=1}^{J} a_{mj}x_j \leq b_m\right] \geq \beta \ m = 1, \ldots, \ M \tag{4}$$

$$E(GM) = \sum_{j=1}^{J} E(cm_j)x_j \tag{5}$$

where

$$a_{nj} (a_{mj}) = Technical \ coefficient \ matrix \ of \ the \ nth \ input \ (mth \ ``non \ certain'' \ input) \ and \ the \ jth \ activity$$

$$\beta = Confidence \ level$$

$$E(cm_j) = Expected \ contribution \ margin \ of \ the \ jth \ activity$$

$$RP = 0.5\rho \sum_{i=1}^{J} \sum_{j=1}^{J} V(cm_i, cm_j)x_ix_j \tag{6}$$

where

$$\rho = Farmer's\ absolute\ risk\ aversion\ coefficient$$
$$V(cm_i, cm_j) = Variance\ covariance\ matrix\ of\ the\ ith\ and\ the\ jth\ activity's\ contribution\ margin \tag{7}$$
$$xj \geq 0\ j = 1, \ldots, m$$

Equation (2) indicates that the risk caused by the fluctuation of the contribution margin of activities—which is caused by the variation in the yield or price—is included in the objective function of the model in the form of the farmer's risk premium. However, according to Equation (6), return fluctuations do not affect all farmers in the same way because they have different risk attitudes, which is reflected in their risk aversion coefficients.

Another aspect of risk captured in this optimization is related to the uncertainty in inputs such as cash at hand and herd size (see Equation (4)). It is assumed that the farmer has certainty about the endowment of some inputs at the beginning of each year (Equation (3)) but is uncertain about others (Equation (4)). These uncertain constraints are specified to be met with a given probability (confidence level) [69,70].

These variables contain risks as their outcomes depend on a combination of several risky factors. To include these risks in the right hand side of the optimization model, the means and standard deviations were calculated from the distribution (i.e., cash at hand and herd size) and included in the chance-constrained programming. Following [71], Equation (4) was entered into the optimization model in the mathematical form of Equation (8):

$$\sum_{j=1}^{J} a_{mj}x_j \leq E(b_m) - \sigma_{b_m}(1 - \beta)^{-0.5} \tag{8}$$

where $E(b_m)$ and $\sigma_{b_m}$ are the expected value and standard deviation of $m$th non-certain input, respectively.

Equation (6) includes a parameter for risk aversion to account for the risk behavior of the farmers. As this parameter is subjective [72,73], the model was solved for different risk aversion coefficients, each of the results were discussed with experts from the region, and the most suitable and efficient production plan was chosen for the farmers [74]. Risk was incorporated into the objective function using a quadratic programming approach [72,75,76]. The standard deviation for the total gross margin was calculated from the variance-covariance matrix of contribution margins for all production activities [68]. The household utility was measured using the discretionary income, which is the income available for household use after paying for all essential expenses, including household consumption, clothing, school fees, etc. [68,77]. The farming household comprises the household head, their spouses, children, or other people in the household that are able to earn any kind of income (see File S3 in the Supplementary Materials for all assumptions and File S4 for the constraints included in the optimization model).

All mathematical equations included in the optimization model are documented in File S4 of the Supplementary Materials. For the chance-constrained programming, we used the General Algebraic Modeling System (GAMS), version 31.2 with the solver DICOPT.

2.5.5. The Integrated Model-Model Coupling

The three models (crop model, farm optimization model, and CLEM model) were recursively integrated on annual basis. As a first step, the crop model simulations were conducted for all crop and fertilization levels for all weather scenarios, scenario members, and years. The simulation results for the crop grain and biomass yields were stored in a database for access by the CLEM and optimization models.

Then, for each farm typology, the CLEM and optimization models were parameterized with initial farm management activities from the survey. Next, starting with one weather scenario, the simulation proceeded by using the CLEM model for the first year of the weather ensembles, which resulted in a distribution of values for the herd size and cash at hand. We ran CLEM for the first year using every ensemble member to obtain 30 independent outputs from the 30 ensemble members. Livestock were only considered



sold if there was not enough money to meet household expenditure, also considering available credit.

The optimization model, in turn, simulates annual crop and land allocation, which are updated in CLEM the following year (i.e., year 2), as shown in Figure 2 above. This process is repeated for 5 years to obtain 30 different outputs from 30 ensemble members. This simulation is also repeated for different risk aversion parameters in the optimization model. This is then repeated for each farm type and weather scenario.

The model results at the end of the 5-year simulation show the responses of the different farms to the various weather conditions, and we observed how the integrated model captures shock, which in this case was due to weather variability. The same simulation was also performed using CLEM without interacting with the optimization model. This was conducted in order to compare the results of the current cropping pattern as simulated by CLEM (which is comparable to many studies in the literature) with the results of the integrated model (which provides a step forward in the possibility of making complete assessments). The integrated model was developed in R software v4.3.0 [48].

### 3. Results

#### *3.1. Farm Typology*

The names given to the derived typology are: low-resource-endowed farms (LRE), medium-resource-endowed (MRE), and high-resource-endowed farms (HRE). The LRE farms, as shown in Table 3, are farms with predominantly female household heads and have relatively smaller land and household sizes. The MRE farms are composed of predominantly middle-aged male household heads, with small household sizes and relatively average land size, while the HRE farms are farms with male household heads, large household sizes and relatively large farm sizes.

**Table 3.** Differences among the three farm types.

| | Unit | LRE * | MRE * | HRE * |
|---|---|---|---|---|
| Adult in household | | 1 | 2 | 2 |
| Children (between 6 and 18) | | 1 | 1 | 5 |
| Children (less than 6) | | 0 | 0 | 2 |
| Remittances | GHS/year | 300 | 338 | 843 |
| Non-farm income | GHS/year | 567 | 1431 | 482 |
| Income from livestock sales | GHS/year | 500 | 441 | 1014 |
| Farm maintenance cost | GHS/year | 150 | 208 | 311 |
| Energy spending cost | GHS/year | 100 | 83 | 170 |
| Household living costs | GHS/year | 120 | 735 | 981 |
| Cash at hand (beginning of the season) | GHS/year | 126 | 1331 | 2393 |
| Average amount of loan | GHS/year | 47 | 1906 | 2536 |
| Loan rate | % per month | 8 | 8 | 8 |
| Input expenses (GHS) | GHS/year | 73 | 663 | 1757 |
| Total land area (hectare) | ha | 0.9 | 4.0 | 6.9 |
| Machinery rental cost (GHS) | GHS/year | 148 | 275 | 462 |
| Cattle | | 12 | 6 | 6 |
| Goat | | 2 | 9 | 8 |
| Sheep | | 0 | 3 | 6 |
| Poultry | | 14 | 19 | 18 |
| Animal supplement costs (GHS) | GHS/year | 12 | 61 | 105 |
| Veterinary visit cost | GHS/year | 0 | 10 | 25 |

* LRE represents low-resource-endowed, MRE represents medium-resourced-endowed and HRE represents high-resource-endowed farms, respectively.

The household size has an impact on farm production as in many cases, they serve as labor for farming activities [78]. On the one hand, this can imply a relatively cheaper source of labor for the HRE farms compared with the LRE and MRE farms. On the other hand, a large household size means that more people in the household have food requirements.

The positive correlation between household size and farm size found in this study is in agreement with the real farms in the study area, as highlighted by [79].

### 3.2. Crop Yield

Figure 3 shows the distribution of crop yields in the bad and good weather scenarios as simulated by SIMPLACE. As expected, the yield from maize with a low fertilizer rate was the lowest among all maize fertilizer rate crops, and the yield was not so different under the two weather scenarios because it is limited more by nutrient deficiency than by climate. Rice is a high-yielding crop in the area as it produces about 4000 kg ha$^{-1}$ on average in good weather. For the cereals, variability in the yields were generally higher in the bad years compared with the good years and increased with the amount of fertilizer applied.

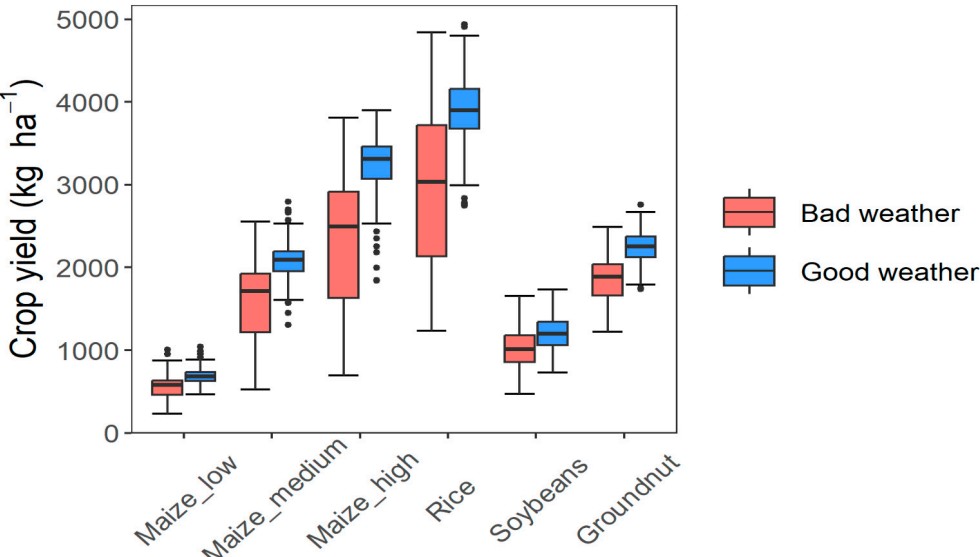

**Figure 3.** Distribution of crop yields for the two weather scenarios (good and bad), red bars show the distribution in 'bad' weather ensemble members, and blue bars show the distribution in 'good' weather ensemble members. The boxplots indicate 25th to 75th percentiles. The black dots represent outliers. The figure is obtained base on the average yield of the 30 ensemble members.

### 3.3. Economic Analysis

The modeled farm income comprises both on-farm and off-farm income, which includes offering labor services to other farmers and selling farm products among other income sources. Maize is mainly cultivated as a food crop in the region, with relatively low gross margins (Table 4) per hectare, especially when household labor costs are accounted for. Rice, on the other hand, is the most economical crop in the region, which explains why the crop has a relatively high gross margin compared with the other crops (Table 4). Table 4 was calculated based on the current production activities of the farms in the study area.

### 3.4. Optimal Land Allocation

The optimal cropping pattern for each farm type under both weather scenarios is presented in Figure 4. For the current cropping activity, both the LRE and MRE farms cultivated a high share of maize-low (28% and 33% of their land area, respectively), which is expected due to the high cost of fertilizers in the study area [80]. The HRE farms, on the other hand, were able to invest in fertilizers to cultivate a high share of maize-medium because they could afford it. However, under the bad weather scenario, all farm types allocated their land to maize-low only, and the proportion of land allocation to maize considerably declined to 5% and 1% for the LRE and MRE farms, respectively. This occurred as a result of it becoming expensive to invest in fertilizers. Although many studies have highlighted that maize yield can be increased through increasing fertilizer application

rates [81,82], Ref. [83] noted that water availability is another major limiting factor of maize yield. In addition, the total land area cultivated by the MRE and HRE farms became much smaller, reducing from 4 ha and 5 ha to 1 ha and 2 ha, respectively, as a result of the poor yield under the bad weather scenario. Farmers were better off allocating a large share of their land to groundnut and soybean under the bad weather scenario. As the weather scenario changed from bad to good, the land share for cash crops increased for all farm types. In addition, under the good weather scenario, the need to diversify the crop choice reduced, and this is evident with the cropping patterns for all the farm types, where over 90% of land area was allocated to rice for LRE farms and more than 80% was allocated to rice and groundnut for MRE farms.

**Table 4.** Average contribution margin per crop per farm (in GHS/ha) based on current production data.

| Cost-Benefit Table with Survey Data | | Maize-Low | Maize-Medium | Maize-High | Soybean | Upland Rice | Groundnut |
|---|---|---|---|---|---|---|---|
| | Tillage | 1.5 | 3.4 | 5.1 | 2.1 | 3.6 | 4.1 |
| | Fertilization | 6.0 | 20.1 | 20.7 | 5.1 | 4.9 | 0.2 |
| | Sowing | 13.4 | 36.1 | 43.7 | 20.9 | 7.9 | 35.1 |
| | Weeding | 13.7 | 39.2 | 56.6 | 25.0 | 52.3 | 39.3 |
| | Harvesting | 16.4 | 50.5 | 58.9 | 40.5 | 55.9 | 54.6 |
| | Threshing | 4.3 | 5.9 | 21.2 | 8.4 | 5.7 | 12.3 |
| | Total | 55.3 | 155.2 | 206.2 | 102.0 | 130.4 | 145.6 |
| | Tillage | 88.3 | 146.7 | 254.8 | 151.5 | 377.4 | 267.5 |
| | Fertilizer + service | 210.4 | 1234.6 | 2165.6 | 209.6 | 680.1 | 34.5 |
| | Seed + service | 19.4 | 15.5 | 57.2 | 128.5 | 111.6 | 113.1 |
| Input cost (cedi per ha) | Herbicide + service | 77.2 | 121.1 | 398.5 | 110.4 | 185.8 | 157.8 |
| | Harvesting | 13.8 | 27.3 | 70.4 | 28.2 | 26.8 | 40.5 |
| | Threshing | 15.2 | 6.5 | 55.4 | 27.2 | 20.8 | 44.1 |
| | Total | 424.3 | 1551.7 | 3001.9 | 655.3 | 1402.4 | 657.5 |
| Total variable cost (cedi per ha) | | 1530.5 | 4654.7 | 7126.5 | 2696.3 | 4011.2 | 3570.4 |
| Average yield (kg per ha) | | 660.6 | 2162.2 | 3294.7 | 1600.9 | 4229.0 | 3037.3 |
| Crop price (cedi/kg) | | 1.7 | 1.7 | 1.7 | 1.8 | 1.5 | 1.7 |
| Total revenue (cedi per ha) | | 1101.0 | 3603.6 | 5491.2 | 2935.0 | 6343.5 | 5062.2 |
| Gross contribution (cedi per ha) | | 676.7 | 2051.9 | 2489.3 | 2279.6 | 4941.1 | 4404.7 |
| Contribution margin (cedi per ha) | | −429.5 | −1051.1 | −1635.3 | 238.7 | 2332.4 | 1491.9 |

These were obtained from the survey and they are used to parameterize the model.

### 3.5. Effects on Incomes and Assets

The probabilities of the increases in assets after five simulation years are shown in Figure 5 for the integrated model with different values of the risk aversion coefficient. The results show that in good weather scenarios and with a low risk aversion coefficient, the probability that farmers would see their income increase over the 5-year simulation period was more than 60%. This is shown in Figure 5A–C, where the risk aversion coefficients used were 0, 0.0001, and 0.001, respectively. This means that farmers' assets will increase and they will be in a better position to cope with shocks if their management decisions are influenced by a model that takes into account the full situation. However, the probability of increasing income was higher under the good weather scenarios; in the bad weather scenarios, these probabilities did not fall below 50% for all farm types. If the farmers continue with their current management practices, they are not able to adjust in response to shocks or other factors because they are not well informed to react to the shocks they may encounter; as a result, they have a lower probability of increasing their income in both the bad and good weather scenarios. In addition, as the risk aversion coefficient increased (Figure 5E–G), there was a much lower probability that the farmers' income would increase after 5 years. This is expected as at a high risk aversion, the farmers' likelihood of perceiving a greater

probability of losses increases [84], and they tend to avoid risky investments, leading to lower incomes [85].

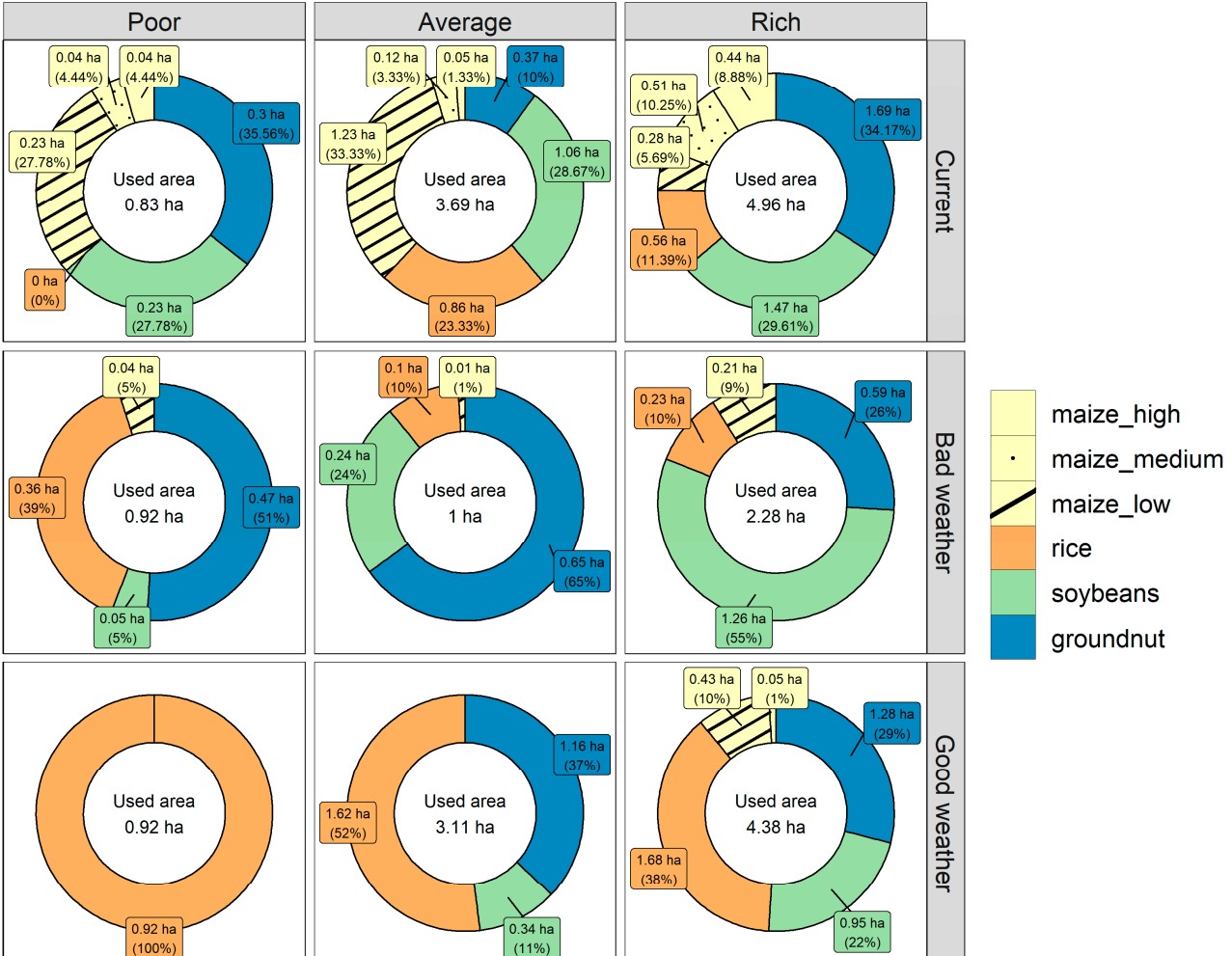

**Figure 4.** Cropping pattern in the different weather scenarios. The middle row shows the optimal cropping pattern in the bad weather scenario, the top row shows the current distribution, and the bottom row shows the distribution in the good weather scenario. The first column represents the low-resource-endowed farms, the second column represents the medium-resource-endowed farms, and the third column represents the high-resource-endowed farms. The yellow color with stripes in the donut plot represents the land allocation to maize with low fertilizer rates, the yellow with dots shows land allocation to maize-medium, the plain yellow color shows the land allocation to maize-high, the green color shows the land allocation to soybeans, the blue color shows the land allocation to groundnut, and the light orange shows the land allocation to rice.

For both the integrated model and CLEM, majority of the livestock were sold at the beginning of the simulation to enable the farm households to meet their consumption needs. In the subsequent years, few livestock were sold. This was expected because during the data collection process and subsequent discussions with experts from the study area, it was observed that farmers usually run out of cash after planting and before harvest. During these periods, they meet their needs by borrowing. The model therefore sold most of the livestock at the beginning of the simulation to cover the households' minimum expenses. This is reflected in Figure 6, where all of the farm households have about a 50% probability of a smaller herd size at the end of 5 years.

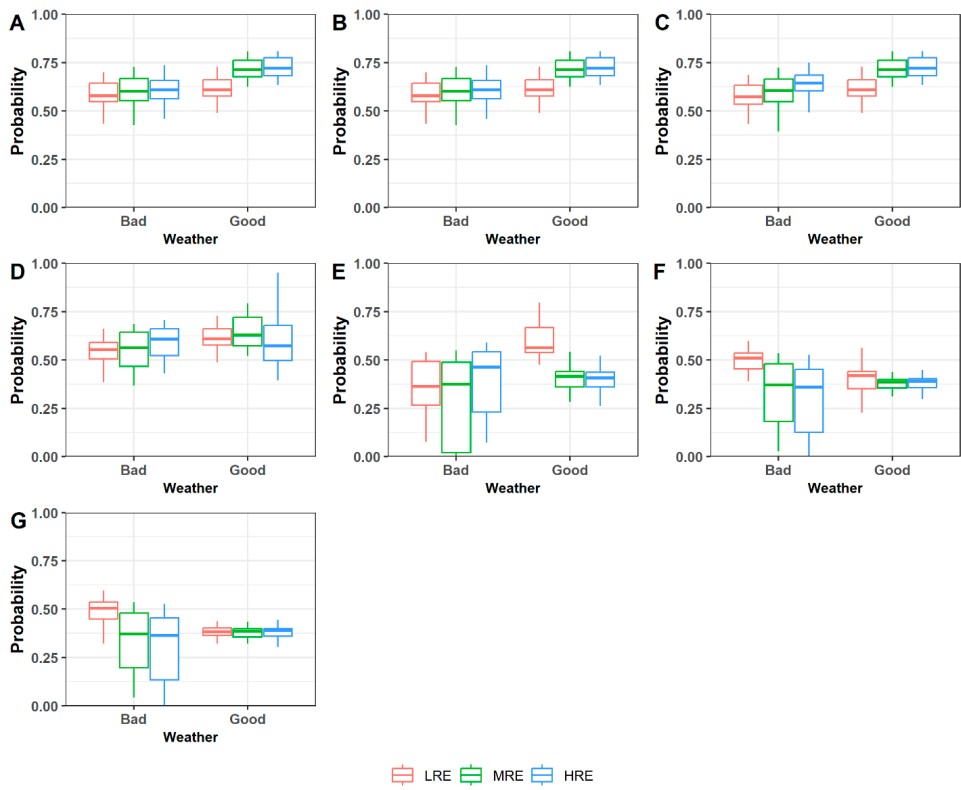

**Figure 5.** Distribution of probability that farmers' income will increase over 5 years, as simulated by the integrated model. These probabilities were obtained by comparing the average income in the first 3 years of the simulation to the last 2 years of the simulation. (**A**)-Simulation result with risk aversion coefficient 0; (**B**)-risk aversion coefficient = 0.0001; (**C**)-risk aversion coefficient = 0.00; (**D**)-risk aversion coefficient = 0.01; (**E**)-risk aversion coefficient = 0.1; (**F**)-risk aversion coefficient = 1, (**G**)-risk aversion coefficient = 4.

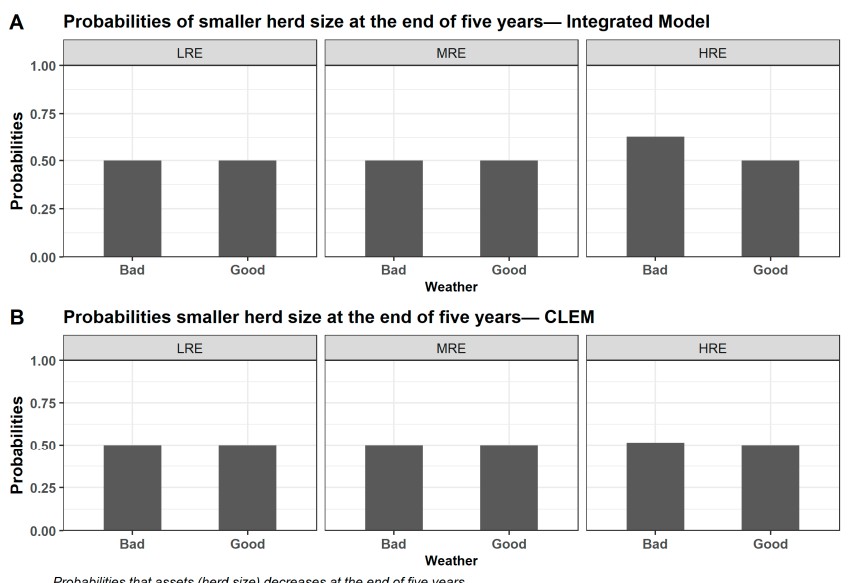

**Figure 6.** Probability of smaller herd size at the end of five years. Results were obtained by comparing the average herd size after accounting for the first initial sales (at the beginning of the simulation) to run the model to the average herd size at the end of the simulation. (**A**)—presents the result from the integrated model, while (**B**)—presents the result from CLEM, which is based on the current cropping patterns of the farmers.

## 4. Discussion

### 4.1. Relevance of the Integrated Model

Given increasingly variable and extreme weather conditions and other shocks (markets, disease, war), supporting the sustainable intensification of farming systems will require assessing how resource allocation decisions are altered and affected by shocks [86] and what the implications are for longer-term developments of assets, including natural capital [87]. Indeed, it is well known that many intensification options and an improved agronomy are associated with larger potential losses during bad weather years [12,21] (see Figure 4). The correct mix of risk reduction, risk transfer, and enabling prudent investments to cope with agronomic risks will differ based on the farm [9,37,88], agro-ecology, market, and institutional context [89]. It is therefore important to develop risk assessment frameworks to understand the appropriate risk management pathways to achieve sustainable intensification. The integrated model presented here offers a novel approach to make such assessments by considering how negative weather could affect production and, in turn, future investments, which was accomplished by combining farm-level optimization and simulation with process-based crop simulation modeling. Like many other bio-economic farm optimization approaches, our optimization model contains the strong assumption that farmers allocate resources and make production decisions to maximize their gross margins as modulated by their risk aversion characteristics. Through linking our optimization model with an annual simulation model (CLEM) that accounts for monthly resource flows, our annual optimization approach explicitly accounts for how bad weather affecting crop yields in the previous season limits cash availability and may alter subsequent cropping system management decisions. Many models have studied the effects of weather and other production risks on different household or farm components such as household production, land degradation [68,90], the farm production system [91], and prices and subsidies [92]. However, unique in our approach is the desire to drive the integrated model with a large ensemble of possible weather realizations (for both good and bad weather scenarios) to assess the probability of a given outcome. With the probabilistic approach, we allow for the assessment of how likely outcomes (changes in incomes, assets, and cropping patterns) are, and this is understood as a good basis for supporting change at local levels [93].

### 4.2. Evaluation of Land Allocation Outputs

Considering the high cost of fertilizers in the study area, it is uneconomical to use fertilizers during extreme weather cases due to the high-cost-and low-benefits of fertilizers during these periods [80,94]. This is reflected in the crop choices in the integrated model, which tends towards the cultivation of crops with low fertilizer rates in the bad weather scenario, as the weather is most likely the yield limiting factor in this case and not the soil fertility [83,95]. For farm households to cope with adverse weather conditions, the model shows that one should allocate a higher proportion of land to groundnut and soybeans. In favorable weather years, i.e., good weather scenarios, it becomes economical to allocate most of the land area to rice and buy food crops such as maize from the market to feed the households; this is particularly the case for LRE farms with land areas of less than 1 ha. These findings are also supported by the profitability analysis report of rice by [96], which indicated that rice is very profitable in normal weather. In addition, [97] concluded that rice yield and profitability significantly increase in good weather scenarios. One main reason why the model is able to allocate a large share of land to economic crops is because we did not include land allocation constraints for food crops. This decision was made because many times the model prioritizes food crops, which are less economical, thereby reducing farmers' income. Instead, the model accounted for consumption costs, which could either come from the farmers' own production or from the market.

### 4.3. Assessing the Probability of Outcomes

Ref. [98] highlighted that after several bad weather years, many farmers experience different kinds of difficulties, including the need for external support to have food. How-

ever, the farming households in our integrated model were able to improve their incomes at the end of 5 years, even in bad weather scenarios. This is possible because the farming decisions are informed by taking weather, farm assets, household consumption and other important farmers' conditions into consideration. In addition, the higher the risk aversion coefficient used for the simulations, the more likely that the farmers' income was reduced at the end of 5 years, because smaller land areas are allocated to crops, the farmers are more likely to diversify their cropping activity and the farmers tend to cultivate more food crops (see Table S2 in the Supplementary Materials). This is the case with several studies that assume that farmers are highly risk-averse as they choose farm plans that provide a satisfactory level of security (e.g., growing larger shares of food crops in the case of smallholder farmers), even if it requires them to sacrifice income on average [33,74,75].

The farmers' herd size, particularly the large ruminants, are a major indication of the farmers' wealth, and they will only sell such livestock as a last resort. This means that even if a farm household has small cash at hand, livestock size is another factor in determining the farmers' wealth [4,22,99]. For the integrated model and the current management systems simulated by the CLEM model, the farmers are able to meet their household expenditure after the first year without selling livestock, thereby preserving their wealth [100,101]. However, farmers sold livestock according to their household size at the beginning of the simulation because they do not have enough resources for all activities. Selling livestock can be prevented if more flexible-term, low-interest rate loans are available to the farmers.

### 4.4. Inclusion of Risk

The risk level is affected by the degree of the variances and the relationship of the covariance of the contribution margin of different enterprises (e.g., different crops on the field) in combination with each other [102]. An advantage of the integrated model over the CLEM model is that the integrated model considers several risk elements. This shows in many ways how combining activities can help to reduce the farmers' risk [76] and that the cropping activities and subsequent income of the farmers are the results that are obtained after considering the risk elements and risk behaviors of farmers. The chance-constrained programming includes the introduction of a confidence level that artificially lowers the probability of resource availability to a certain limit, which allows for the decision-maker to have a certain level of probability that this limit will be met [74]; in other words, it is the level of certainty that the decision-maker has that the constraint will be met. For the chance-constrained model, we applied a 95% confidence level. This implies that the constraints of the model will be satisfied with a probability of 95%.

### 4.5. Study Limitations

One of the major limitations of this study is the assumption of constant costs. We adopted this approach because the weather data used are simulated data and not actual observed historical data. Ideally, adding yearly variation to input costs would better justify the results of this study. In addition, we did not account for payment for household labor. As this is an ideal situation for economic studies, in this case, the household expenditure and consumption were accounted for, but paying the household labor the prevailing wage rate would make most of the farmers go bankrupt in many cases.

Another limitation of this study is that we did not consider long-term crop rotation effects such as soil organic carbon (SOC) on soil characteristics. We neglected the rotation effects because the simulations in this study were carried out over a relatively short-term period (5 years). In a future study, it would be interesting to see the long-term trajectory of SOC under different crop rotations and include such changes in the integrated model to optimize cropping systems, considering not only economic aspects but also environmental aspects.

## 5. Conclusions

This study presented an integrated bio-economic model that simulates the impact of weather on farm management. The main focus of this paper was to compare the results of the integrated model with simulations from a household model (CLEM). The model offers a novel approach for risk assessment frameworks, which can help to understand risk management strategies and pathways and achieve sustainable intensification. Additionally, the model helps farmers to adapt to climate change by making informed management decisions with the long-term effect of increasing their income. By using a large ensemble climate forcing dataset, the model is able to assess the probabilities of outcomes. The conclusion from this study is that the integrated model provides more support for smallholder farmers under different weather conditions as the farm-level resource allocations are informed by environmental conditions, resource availability, and farmers' risk perceptions. An interesting finding from this study is that farmers are advised to allocate a large portion of their land to cultivate rice crops, particularly in scenarios of good weather. By doing so, they have enough money to meet their household expenditure until the following harvest seasons. With the results of this study, farmers can therefore shift their focus from cultivating a large proportion of food crops to more market-oriented crops as they become better off, and they are thus able to cope much better under bad weather scenarios.

To better understand what kind of management options can support farmers' investments in sustainable intensification options and bring them out of the poverty trap, it is desirable to carry out a future study that can test different risk management options using our integrated modeling framework. This can be accomplished by testing different risk management strategies, such as insurance options, in combination with sustainable intensification options, such as residue management. Such a study can reliably inform farmers of the best risk management options for them under different weather scenarios.

**Supplementary Materials:** The following supporting information can be downloaded at: https://www.mdpi.com/article/10.3390/su15097386/s1, File S1: Calculations for classifying maize crop with varying fertilizer application intensity; File S2: CLEM model assumptions; File S3: Farm optimization model assumptions; File S4: Optimization model constraints; Table S1: Description of mathematical symbols used in the optimization model; Table S2: Scenario analysis of cropping activity under changing risk aversion coefficient.

**Author Contributions:** Conceptualization, O.O.A., H.W., P.Z. and J.S.; methodology, O.O.A., Y.-U.K., H.W., P.Z., J.S., S.-A.H.-Y., D.S.M., A.L.A., K.v.d.W., P.C.S.T. and S.G.K.A.; software, O.O.A. and Y.-U.K.; validation, H.W., P.Z., J.S., S.-A.H.-Y., D.S.M., A.L.A., K.v.d.W., P.C.S.T. and S.G.K.A.; data curation, O.O.A. and Y.-U.K.; writing—original draft preparation, O.O.A.; writing—review and editing, O.O.A.; visualization, O.O.A.; supervision, H.W.; project administration, H.W.; funding acquisition, H.W. All authors have read and agreed to the published version of the manuscript.

**Funding:** This research received no external funding.

**Institutional Review Board Statement:** Not applicable.

**Informed Consent Statement:** Not applicable.

**Conflicts of Interest:** The authors declare no conflict of interest.

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
