# Peer review of "Accounting for Weather Variability in Farm Management Resource Allocation in Northern Ghana: An Integrated Modeling Approach"

_sustainability, doi:10.3390/su15097386_

Round 1
Reviewer 1 Report
Modeling the distribution of resources in the agricultural economy in the context of the variability of nature is relevant, but not new. This problem is studied by scientists for a long time. However, the work is interesting and important. The authors confirmed that the modeling of agricultural production of the farmer is an important condition for its sustainable development.
The literature review refers to the central issue of the paper, it is quite extensive, relevant and thorough. The review will be of interest to other researchers. I would like to mention that the authors have comprehensively studied the literature on the issue published over the last five years. References are correct. The structure of the article meets the requirements. The results complete previous results on the matter and are supported by references.
However, there is a small comment to the work.
1. Annotation must be reduced to 200 words.
2. In the annotation, we do not see the specific results of the model. Efficiency of the results. Why did this model be considered?
3. In the conclusions, it is necessary to show the novelty and practical significance of the results obtained, as well as indicate the main directions for further research in this area.
The article can be adopted after correcting the comments.
Reviewer 2 Report
In their manuscript, the authors present a novel method for evaluating an integrated bio-economic model. Using a large ensemble of climate modeling data, this was created to simulate the outcomes of resource allocation based on optimization of gross margins at the farm level in response to weather and management. Using data from the EC-Earth global climate model, the utilized large ensemble dataset contains 2000 years of current weather conditions. The integrated bio-economic model was constructed with a process-based crop model (SIMPLACE framework), a farm simulation model, and a developed optimization model as its tripodal bases. Utilizing a large ensemble climate forcing dataset that permits the evaluation of the probabilities of outcomes represents the state of the art in this research. This significant benchmark establishes this as a new leading implementation for climate change, resource allocations, and farmers' risk perceptions.
I find the manuscript to be generally well-written, and in my opinion, the manuscript is suitable for publication in sustainability.
There are specific comments in the manuscript.
For instance, "developed by [39] I propose that this be "developed by CGIAR CASCAID."
Reviewer 3 Report
The article is very interesting to me, I conducted surveys myself and I know how difficult it is to prepare surveys so that farmers can answer them without much difficulty. In addition, all answers and data must be evaluated and verified, and appropriate programs must be used for the analysis so that the results are objective. All this is included in this article, the introduction is correctly and interestingly written, the methodology is clear and correctly written, and the results are well discussed and interpreted.
For me, as a European, the country and agriculture and the main crops (cash crops) and animals in this country are interesting. I had a student from Ghana who partly introduced me to the problems related to agriculture in this country, while the article presents this branch of the economy in this country even more, taking into account the impact of climate change.
Author Response
Thank you. There are no comments to respond to here.